# “Bring Your Own Device”—A New Approach to Wearable Outcome Assessment in Trauma

**DOI:** 10.3390/medicina59020403

**Published:** 2023-02-19

**Authors:** Benedikt J. Braun, Tina Histing, Maximilian M. Menger, Julian Platte, Bernd Grimm, Andrew M. Hanflik, Peter H. Richter, Sureshan Sivananthan, Seth R. Yarboro, Boyko Gueorguiev, Dmitry Pokhvashchev, Meir T. Marmor

**Affiliations:** 1Department of Trauma and Reconstructive Surgery, Eberhard-Karls-University Tuebingen, BG Unfallklinik, 72076 Tuebingen, Germany; 2Human Motion, Orthopaedics, Sports Medicine and Digital Methods Group, Department Precision Health, Luxembourg Institute of Health, 1445 Luxembourg, Luxembourg; 3Department of Orthopaedic Surgery, Southern California Permanente Medical Group, Downey Medical Center, Kaiser Permanente Downey Medical Centre, Downey, CA 90242, USA; 4Department of Orthopaedic Surgery, University of Ulm, 89081 Ulm, Germany; 5ALTY Orthopaedic Hospital, Kuala Lumpur 50450, Malaysia; 6Department of Orthopaedic Surgery, Division of Adult Orthopaedic Trauma, University of Virginia, Charlottesville, VA 22903, USA; 7AO Research Institute Davos, 7270 Davos, Switzerland; 8Orthopaedic Trauma Institute (OTI), San Francisco General Hospital, University of California, San Francisco, CA 94720, USA

**Keywords:** digital outcome assessment, fracture, traumatology

## Abstract

*Background and Objectives*: Outcome data from wearable devices are increasingly used in both research and clinics. Traditionally, a dedicated device is chosen for a given study or clinical application to collect outcome data as soon as the patient is included in a study or undergoes a procedure. The current study introduces a new measurement strategy, whereby patients’ own devices are utilized, allowing for both a pre-injury baseline measure and ability to show achievable results. *Materials and Methods*: Patients with a pre-existing musculoskeletal injury of the upper and lower extremity were included in this exploratory, proof-of-concept study. They were followed up for a minimum of 6 weeks after injury, and their wearable outcome data (from a smartphone and/or a body-worn sensor) were continuously acquired during this period. A descriptive analysis of the screening characteristics and the observed and achievable outcome patterns was performed. *Results*: A total of 432 patients was continuously screened for the study, and their screening was analyzed. The highest success rate for successful inclusion was in younger patients. Forty-eight patients were included in the analysis. The most prevalent outcome was step count. Three distinctive activity data patterns were observed: patients recovering, patients with slow or no recovery, and patients needing additional measures to determine treatment outcomes. *Conclusions*: Measuring outcomes in trauma patients with the Bring Your Own Device (BYOD) strategy is feasible. With this approach, patients were able to provide continuous activity data without any dedicated equipment given to them. The measurement technique is especially suited to particular patient groups. Our study’s screening log and inclusion characteristics can help inform future studies wishing to employ the BYOD design.

## 1. Introduction

The use of wearable activity monitors is steadily increasing worldwide [1]. Over the recent 10 years, this general trend is reflected in medicine by a variety of publications on activity tracking via wearable systems with objective outcome evaluation. To date, Patient-Reported Outcome Measures (PROMs) are widely accepted to assess both functional outcome and health status following an orthopedic trauma injury. However, despite their advantages in generating patient-reported outcomes for comparisons, these scoring tools suffer certain biases, influenced by the patient’s cooperation and compliance, quality of reply, and recall [2,3]. Especially during the evaluation of the pre-injury physical function and patient’s condition, recall bias can significantly influence the reliability of obtained scores [4]. Furthermore, all scoring systems are subjected to timepoints and intervals for recording of the corresponding data, making daily assessments over longer periods of time not feasible. Thus, objectively measuring a patient’s outcome with wearable technology is an evolving field in orthopedic trauma surgery, with increasing interest for research and clinical purposes [5,6,7].

A recent review of the existing biomedical literature between 2010 and 2019 in PubMed and Embase—related to orthopedic trauma surgery and outcome assessment with wearable activity monitors—has confirmed the increasing presence of these systems [7]. The authors identified one hundred and thirty-six relevant publications with an increasing number of reports per year. The most commonly applied wearable technologies were accelerometry, followed by plantar pressure insoles. Accordingly, plantar placement was one of the most common locations for application of wearable sensors, followed by the waist and extremities. Hip and fragility fractures were generally the most studied fracture types, followed by other lower extremity fractures. Commonly reported outcome metrics were step count, activity time, and sleep duration. However, despite this increasing trend of using wearable activity monitors, no clear standards or best practice guidelines regarding the optimal outcome metric, technology, and measurement technique have been established so far [7,8].

A recent survey among AO Trauma surgeons has confirmed that wearable outcome measures are gaining popularity not only in the research field, but also to monitor patient activity and recovery in a clinical setting. Almost 20% of the surveyed surgeons were already using wearable systems as part of their clinical treatment pathway. Apart from wireless heart rate and oxygen saturation measurements, the most prevalently employed wearable technology was accelerometry in conjunction with smartphones (75.4%). The most commonly measured wearable activity metric was general patient activity. Interestingly, almost 20% of the surveyed surgeons were still unsure about the selection of the best measurement technique to obtain meaningful outcome data [6].

Both the clinical and review data have demonstrated that the prevalent measurement strategy for wearable activity monitoring considers patients equipped with dedicated wearable devices (i.e., smartwatches/bands), which are selected by the clinician or researcher and then used by the patient (DD strategy). This allows for generation of a uniform dataset for comparisons within one study focusing on activities according to the inclusion criteria post injury [7,9]. Another potential strategy is to use devices for performance measurement that the patient already has (i.e., “Bring Your Own Device” or BYOD strategy). The BYOD strategy has the potential to increase compliance and obtain pre-injury activity data [1]. However, to our knowledge, no orthopedic trauma studies exist on the BYOD strategy, or such comparing between the BYOD and DD strategies. This gap in the medical literature poses an open research question of high relevance to researchers and clinicians in the field of traumatology [5,6].

Therefore, the aim of the current study is to introduce proof-of-concept clinical outcome data for the BYOD measurement strategy based on data obtained with wearables already used by the patient before injury. Screening and inclusion characteristics, as well as obtainable outcomes, are reported, especially concerning a pre-injury baseline measure.

## 2. Materials and Methods

The study was conducted in accordance with the Declaration of Helsinki and approved by the Ethics Committees of the University of Tuebingen and the University of California San Francisco (UCSF: Protocol code 20-30783, date of approval 22 July 2020; BG: Protocol code 790/2020BO2, date of approval 18 November 2020). Informed consent was obtained from all subjects involved in the study.

Patients were screened for study inclusion in two centers: a University of California San Francisco fracture clinic (UCSF) and the BG Unfallklinik, University Hospital Tuebingen, on behalf of the Eberhard-Karls-University, Tuebingen (BG).

At UCSF, all patients presented to the clinic were screened for eligibility. The inclusion criteria included a recent fracture treated surgically and a planned follow-up at the clinic for a period of six months after the surgery. Additionally, the patients had to be aged 18 years or older; and they had to own at least a personal smartphone/wearable that they knew how to operate. The exclusion criteria included non-English speakers and patients with mental health issues who could not provide consent for medical procedures.

At BG, patients were screened during their in-patient surgical stay or within three weeks after the surgery or the injury event as per availability. All patients over the age of 18 years with a pre-existing wearable (smartphone or body-worn sensor) and any musculoskeletal injury of the upper and/or lower extremity were included. Patients who were unable to provide informed consent, abused drugs, were pregnant, or enrolled in another clinical study were excluded.

Only patients with a minimum of 6 weeks of wearable outcome data were analyzed for the current analysis. Patient age, sex, injured extremity, and all available activity data were recorded both pre and post injury. In addition, descriptive statistics of the UCSF screening log, and UCSF and BG treatment data was performed. Wearable outcome data were plotted for 2-week, 6-week and 3-month follow-up intervals. For the first 6 weeks, all values were continuously normalized to the average activity carried out 7 days before the injury with GraphPad Prism software (GraphPad Prism Version 9; GraphPad Software Inc., San Diego, CA, USA).

## 3. Results

Overall, 48 patients were included in this preliminary, prospective, proof-of-concept study. The average patient age was 45.9 years (range 21–67 years); 13 female and 44 male patients were included. Twelve patients suffered from upper extremity injuries, thirty-three from lower extremity injuries, seven from injuries of the pelvis, and five from combined injuries.

### 3.1. Screening Characteristics

Continuous inclusion was performed at UCSF, and a screening log was kept, noting the reasons for study exclusion. Overall, 432 Patients were screened, of which 27 agreed to be included in the study (6%). Data from 18 patients were available for the final analysis. The age of the patients included in the study was significantly different from those excluded from it (38.6 ± 15.2 years versus 55.6 ± 22.5 years, *p* = 0.0187); 8% of all the screened patients did not have a smartphone available or did not know how to use one. In addition, the age in this group was significantly higher than in all other screened patients (72.1 ± 18.9 years versus 54 ± 22.3 years, *p* = 0.0028). Another 11% of all the screened patients refused to participate. All other exclusions (75%) were due to the inclusion and exclusion criteria of the study, being unrelated to the availability of a wearable device or willingness to use one (Figure 1). Of the patients with available smartphones, 56% were already collecting wearable activity data before the injury event.

### 3.2. Wearable Outcome Data

The most commonly used technology brought by the patients to the study was based on smartphones (100%), followed by wrist-worn wearable technology in conjunction with smartphones (*n* = 9; 18.8%). Of the used systems, 70.8% (*n* = 34) were based on Apple technology (Apple Inc, Cupertino, CA, USA), and 29.2% (*n* = 14) were based on Google technology (Google LLC, Mountain View, CA, USA). Step count was the only metric provided by all wearable systems. The outcome data of this parameter were longitudinally mapped for all patients, and daily averages were calculated at the 2-week, 6-week, and 3-month follow-up timepoints for the parameter step count (Figure 2).

When including only patients with pre-injury data available (*n* = 30), the complete recovery could be tracked and normalized to the daily averages of the available activity outcome parameters for all patients prior to the injury. Data were capped at 100%. This was performed for both step count (Figure 3a), as well as for the other most common metrics provided (Figure 3b). For Apple users, the second most common value provided was “distance”, whereas for Android-based systems it was “cardiopoints”. Three distinct recovery patterns were seen. Furthermore, the timepoint of recovery in patients recovering during the study could be shown as early as during the first 6 weeks after injury. These patterns were irrespective of upper or lower extremity injury and spine or pelvic injury location.

## 4. Discussion

This work is the first to present the feasibility of utilizing a patient’s own wearable activity monitor to track outcomes in orthopedic trauma surgery, introducing a new measurement strategy to orthopedic traumatology: “Bring Your Own Device”. Interestingly, despite a low-powered, proof-of-concept design, different outcome patterns could be observed in this group of trauma patients. Additionally, the measurement strategy can differentiate between patients on the path to recovery and patients not recovering in a timely fashion. Of note, only 19% of the patients refused to participate during screening due to reasons related to the availability of a wearable device or hesitation towards sharing activity data.

Concerning the pre-study enrollment characteristics, the age of patients refusing or unable to participate in the study due to smartphone-related issues was significantly higher than that of the active study participants. Despite the increasing use of wearables by the general public, as well as in clinics and for research purposes [1,6,7], increasing age is associated with less availability and a limited use of smartphones and wearable systems [10,11]. Concerning the feasibility of the bring-your-own-device measurement strategy, our results confirm that this is an influential factor in determining study participation. For the time being, clinicians and investigators should consider employing this strategy in fracture cases predominantly encountered in younger patients, or expect higher exclusion rates as seen in our screening log. However, the increasing use of wearables in the aging population and in the coming-of-age current wearable users will likely reduce this limitation in the future. General statistics on the distribution of wearable systems in the United States population already reveal a trend of their increasing use over the recent 6 years in both age groups of 50 to 64 years and over 65 years. Current studies, specifically investigating the distribution characteristics of smartphones, tablets, and medical applications, have already demonstrated higher distribution rates among the aging and elderly populations [12]. The ownership rates of smartphones according to the United States Health Information National Trends Survey were reported to be over 70% in the age group of 56 to 65 years and approximately 60% in the age group of 66 to 75 years. Only in the population aged over 75 years a significant rate drop below 40% was observed. Accordingly, while almost 50% of the younger participants answered that the smartphone had already helped in tracking their health, only 20% reported this use in the older population. This is certainly comparable to the availability characteristics observed in our study. Generally, the age of the included patients was significantly younger compared with excluded patients. Furthermore, the average age of patients not owning or unable to use a smartphone or another wearable device was considerably higher, being just over 72 years. Research projects aiming at increasing the use of wearable devices and interactions among the elderly population have already reported promising results [11]. In addition, training programs enhancing the competency of smartphone use in the elderly have demonstrated positive effects [13]. This trend towards an increased use of wearable technologies to monitor different health conditions in the aging population will likely increase in various fields over the coming years [14,15]. Apart from the age-related availability, no other meaningful trends could be observed from our enrollment statistics.

For all included patients, the smartphone was by far the most available outcome measurement among the wearables, followed by wrist-worn wearable devices. This is in seeming contrast to the current literature in orthopedic trauma surgery, where the majority of wearable activity metrics are tracked via accelerometry or mobile gait analysis technologies [7]. The latter is a predominant effect of the design of the studies, where the rate of dedicated devices used to measure activity and distribution according to medical-grade systems is quite high, as opposed to dedicated smartphone applications. The use of smartphones and wrist-worn wearable devices in the current study was in accordance with a recent survey analysis performed by the AO Foundation among orthopedic trauma surgeons focusing on current and planned clinical applications of wearable systems. It has been concluded that wearable outcome data were increasingly implemented in the treatment of patients with musculoskeletal injuries, and that the most widely distributed devices to measure general patient activity were smartphones and accelerometry-based wearable systems [6]. However, the technologies and applications employed by the survey participants revealed a high diversity in responses, confirming that clear standards regarding measurement strategies have yet to be defined.

Most commonly, smartphones and wearable devices are used with dedicated apps to measure physical activity and well-being in patients and healthy individuals [16,17,18,19]. This technology allows for various measurement times and outcomes and is considered equally or more reliable than some commercially available wrist-worn systems [20,21]. By selecting a specific wearable device or app that is consistently used within a study, or at a clinical center, comparable outcome datasets are generated, while limiting the study to post-inclusion data [7,9]. Furthermore, when using dedicated devices in addition to personal devices, a patient might have compliance issues, commonly seen when using additional systems [22,23,24]. This additional compliance issue is theoretically reduced by using a wearable device which is already implemented in the patient’s daily routine. However, an issue could exist when the collection of wearable device data is influenced by the injury (e.g., when a wrist-worn wearable device is not worn in radius fracture patients). Assuming that the characteristics of smartphones during activity measurements are not changed by most injuries, the standard error would remain comparable pre and post injury. By normalizing the individual patient recovery data to the pre-injury activity state and assuming a non-considerable change in use by the patient, compliance would not affect the BYOD measurement strategy as could be the case in a dedicated device study. A limited survey analysis showing the comparable use of smartphones in survey participants with or without traumatic brain injury supports this hypothesis [25]. Ultimately, comparative studies of both measurement strategies are necessary to investigate this effect.

A key element of the BYOD strategy is that pre-injury patient activity data is available to the clinician and researcher. This is not possible by wearable activity outcome measurement strategies that assign a dedicated device to the patient upon study inclusion. Our screening characteristics show that roughly half of the patients (56%) were measuring “pre-injury” activity parameters. In these patients, it is possible to track their rehabilitation process prospectively and to also track their return to their previous physical activity state. Our results show the feasibility of this preclinical “activity biopsy”. A return to the pre-injury activity status as early as during the first 6 weeks post injury could be shown. Referencing pre-injury data could allow identification of inadequate recovery and identify necessity of early intervention. Likewise, patients with continuously high wearable activity values could receive additional monitoring measures to track their rehabilitation. Routine clinical measures or score-based outcomes are less likely to reliably assess a patient’s recovery trajectory. Even established patient-reported outcome measures are limited in their capacity to objectively assess pre-injury patient status and thus track the recovery process back to a norm value for the patient [4,26]. These scoring systems are mainly limited by recall bias and are highly dependent on patient participation and answering characteristics; thus, the reported outcome is either tracked prospectively without having an available patient specific baseline, or by comparing it to established norm values with all associated limitations.

The only activity outcome available over all platforms in our study was the daily step count. This metric has been shown in a recent review of orthopedic trauma literature as a primary outcome parameter in fracture studies using wearable systems. It has been further described as a relevant parameter to track neurologic and pneumatological disease progression and is associated with all-cause mortality [7,27,28,29]. The distinct value of this parameter to determine fracture healing has yet to be fully understood; its broad availability and use in wearable research literature does, however, ensure comparability in future research efforts. Despite measurement differences between different operating systems and devices, cross-platform effects are minimized by normalizing the activity outcome data to a pre-injury baseline. Furthermore, centralized data collection apps offering cross-platform conversion are becoming increasingly available and will allow comparisons of absolute values across operating systems and devices in future studies [30]. Interestingly, while the other collected outcome parameters differed between the operating systems of Apple and Google, the observed “return to normal function” curves provided similar shapes to each other and also to the step count when normalized. In the early post-trauma phase and proof-of-concept setting, these activity outcome parameters seemed comparably feasible for tracking of the recovery process. The true effect of different hardware and operating systems on the results obtained with the BYOD strategy still has to be determined in future studies focusing on this specific aspect. Certainly, comparability between operating systems, or even phones from different generations by the same manufacturer, is limited, mainly due to differences in the operating software versions installed or different hardware used by one manufacturer in different-generation wearables. Differences in this regard will decrease comparability between patients using different devices to track their activity recovery process. As previously discussed, a number of data-harvesting solutions already exist, which try to account for these variations based on large amounts of wearable data already collected. Ultimately, this limitation is also an advantage of the technique, as the analysis of the recovery process and activity is based on the individual data stream, including the pre-injury wearable activity data. By employing the BYOD strategy, the between-patient comparability is limited as opposed to traditional dedicated device studies, but the within-patient comparability is enabled, allowing for a complete tracking of the patient recovery process in reference to the pre-injury performance.

The study has several limitations due partly to its preliminary, proof-of-concept design. The overall patient number was low, especially considering the heterogenous inclusion of musculoskeletal injuries of the upper and lower extremities, and pelvis. However, despite these limitations, a first proof of concept for the introduced measurement strategy is provided, and meaningful outcome patterns, irrespective of the patient’s injury, are described. Another limitation lies in the heterogeneity of the systems used, as well as the outcome parameters provided within one study. This limitation is inherent to the introduced measurement strategy that relies on whatever wearables patients bring to the study. Of note, step count—a scientifically and clinically relevant activity parameter [3]—was available across platforms. However, this limitation also represents an advantage of the measuring technique, as the analysis of both the recovery process and activity is based on the individual data stream, thus ensuring continuity and comparability to the pre-injury data. This would not be possible when using a dedicated device only handed out after the injury or upon the start of the study. That is why it is the advantage and limitation of the technique that needs to be accounted for. Further studies, having already been set up, will have to determine the effects of using different devices and weigh the disadvantages against the advantages of being able to collect pre-injury patient data for recovery tracking. Finally, this investigation was set up as a feasibility study of the measurement technique at two centers in different regions—one recruiting continuously and one recruiting by availability. This certainly introduces a selection bias to the study. Accordingly, the availability of patients and devices for analysis has only been reported for the study center with continuous enrollment. The low enrollment rate reflects the strict application of the inclusion criteria focusing only on patients with operative therapy and fluency in English. Of note, only a small percentage of patients refused to participate due to the unavailability of, or their unwillingness to use, a smartphone. Fracture-entity-specific studies with continuous enrollment will have to further investigate the general availability of pre-injury activity data in different medical systems, as the inclusion and screening characteristics, activity measures, wearable systems, and availability of pre-injury activity data could have been influenced by the geographic location and regional specifics of the two study centers in Germany and the USA. Despite these limitations, the aim of our study was to demonstrate the general proof of concept to obtain objective, individual pre-injury, and clinical outcome activity data from wearable devices already used by the patients during their everyday life.

## 5. Conclusions

With the “Bring Your Own Device” strategy, providing continuous, daily outcome data in orthopedic trauma patients is feasible. Its major advantage is that patients can use their own device and that pre-injury activity baseline data can be used to assess the individual patient recovery process. The technique is currently more suitable for younger patients due to the age-related distribution characteristics of wearable devices and smartphones; however, the growing distribution of digital technology among the elderly, as well as existing dedicated training programs, will likely increase the useability of this approach in all age groups. Further research is needed to determine the advantages of this new strategy compared to traditional outcome measurement techniques and those employing a dedicated wearable device. Specifically, questions about the effects of using different devices, software, and outcome metrics within one study need to be addressed.

## Figures and Tables

**Figure 1 medicina-59-00403-f001:**
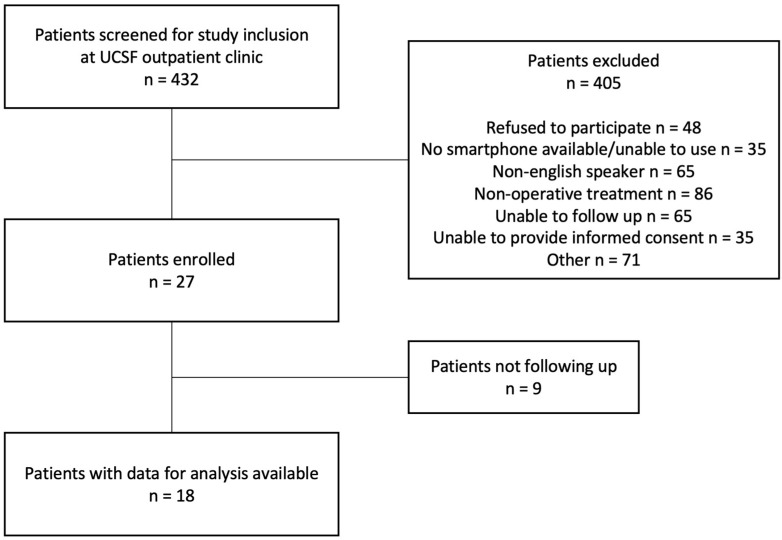
Study flow chart presenting number of patients who were screened, enrolled, and available for final data analysis, together with reasons for exclusion at UCSF from the continuous screening log.

**Figure 2 medicina-59-00403-f002:**
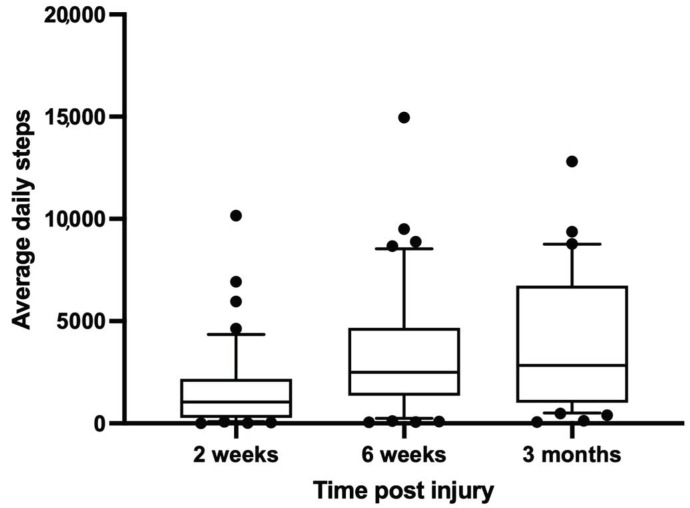
Patient average step count during the injury recovery process at 2 weeks, 6 weeks and 3 months post injury. The *X*-axis shows the time from injury in weeks, while the *Y*-axis shows the average step count per day. Boxes show 1st and 3rd quartile with median, and antennae show 10th and 90th percentile; outliers are identified by dots.

**Figure 3 medicina-59-00403-f003:**
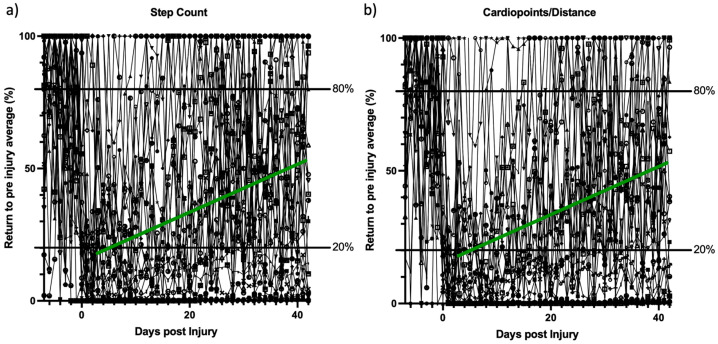
(**a**) shows the pre- and post-injury patient recovery process, as continuously tracked by the daily step-count over 6 weeks. (**b**) shows the pre- and post-injury patient recovery process, as continuously tracked by the most common other available smartphone activity outcomes (Apple: Distance; Android: Cardiopoints) over 6 weeks. All values were normalized to the average individual activity 7 days prior to injury and capped at 100% for better visualization. The *X*-axis shows the time in days (0 = day of injury), while the *Y*-axis shows the normalized activity metric. The horizontal lines show the lower 20% and upper 80% border. The green line represents a drawn trend line of patients recovering during the first 6 weeks.

## Data Availability

All original data are shown in the article. No further data are available.

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
