# Peer review of "“Bring Your Own Device”—A New Approach to Wearable Outcome Assessment in Trauma"

_medicina, 2023, doi:10.3390/medicina59020403_

Round 1
Reviewer 1 Report
This is a preliminary, prospective, proof-of-concept study looking at a new measurement strategy whereby the patient’s own devices are utilized, allowing for both a pre-injury baseline measure and the ability to show achievable results. The authors found that measuring outcomes in trauma patients with the Bring Your Own Device (BYOD) strategy is feasible. With this approach, patients were able to provide continuous activity data without dedicated equipment given to them. This topic is interesting, and the method is useful.
The problems are the following:
1) There is no flow chart to show including patients, and the excluding rate was so high. The serious selected bias may exist;
2) This paper is not a complete article, just like a mini-report;
3) There is no process of data analysis;
4) what was the association between BYOD data and the outcome in trauma patients? This section should be the main result. However, I cannot find the context.
5) the author did not tell how to use BYOD in clinical.
Author Response
Dear Reviewer,
Thank you for providing this thorough review that we feel has greatly improved the quality of the manuscript. We have addressed all comments especially focusing on clarity and understandability of the methods and results in light of the proof-of-concept nature of this study.
Point-by point-answers to all comments are provided below.
We appreciate this opportunity and thank you.
General comments:
- This is a preliminary, prospective, proof-of-concept study looking at a new measurement strategy whereby the patient’s own devices are utilized, allowing for both a pre-injury baseline measure and the ability to show achievable results. The authors found that measuring outcomes in trauma patients with the Bring Your Own Device (BYOD) strategy is feasible. With this approach, patients were able to provide continuous activity data without dedicated equipment given to them. This topic is interesting, and the method is useful.
Reply: Thank you for these kind remarks. We agree that this topic is highly interesting and will be extensively utilized in future studies. Larger scale studies based on the proof-of-concept study results presented here are already planned and funded.
Specific comments:
- There is no flow chart to show including patients, and the excluding rate was so high. The serious selected bias may exist;
Reply: This study was set up as a feasibility study of the measurement technique at two centers, one recruiting continuously and one recruiting by availability. This certainly introduces selection bias, but does not take away from the obtainable data feasibility - which was the aim of this study. We have now better discussed this bias as part of the limitations section. In addition, we have now included a flow chart for the study center that had continuous enrollment and better discussed the high exclusion rate.
- This paper is not a complete article, just like a mini-report;
Reply: Thank you,we agree, the intention of this article was to give a concise report on the feasibility of this new and interesting technique that has to be surely investigated in other larger-scale and entity-focused studies. We extended the manuscript body and discussion to provide more background information to the interested reader. The limitations are now more clearly reflected in a concise report in the according manuscript section.
- There is no process of data analysis; and 4) what was the association between BYOD data and the outcome in trauma patients? This section should be the main result. However, I cannot find the context.
Reply: We agree with the reviewer's comments that this article is lacking clinical correlation of the results. The presented work was set up as a first proof-of-concept for the newly introduced measurement strategy, not focusing on specific fracture entities and their outcome, hence the rather diverse and unstructured inclusion. For this first analysis we wanted to focus on the method itself. With the limited patient number recruited for this feasibility study, the reporting of outcomes in this rather heterogenous group did not seem meaningful.
Presentations of this data at international Trauma and AO Foundation meetings have confirmed the impact this new measurement strategy can have on objective outcome measurement. It is already planned as a measurement technique in larger cohort studies with focused, entity-specific inclusion. This article can then serve as a citeable feasibility study. Hence, the design and method focused on assessment and reporting.
We can fully understand the reviewers concerns and will address these concerns as part of the future design of studies based on the presented data.
- the author did not tell how to use BYOD in clinical.
Reply: Thank you for this comment. We have revised the manuscript to give a better perspective on the clinical, as well as research used scenarios for this technique as part of the adapted discussion section of the manuscript.
Reviewer 2 Report
This is an article on trauma assessment methods for wearable devices. This topic is novel and worthy of investigation. But there are still several issues that need to be discussed.
1. Due to young people's higher acceptance of smart phones and other reasons, the study is more inclined to target young people. The authors are also aware of this limitation in the article.
2. Although the operating systems of Apple and Google provided similar curved shapes to each other, it is still recommended to use the same equipment and system for testing. The authors also do not go far enough to prove whether Apple and Google are similar in terms of how they operate.
Author Response
Dear Reviewer,
Thank you for providing this thorough review that we feel has greatly improved the quality of the manuscript. We have addressed all comments especially focusing on clarity and understandability of the methods and results in light of the proof-of-concept nature of this study.
Point-by-point answers to all comments are provided below.
We appreciate this opportunity and thank you.
General comments:
- This is an article on trauma assessment methods for wearable devices. This topic is novel and worthy of investigation. But there are still several issues that need to be discussed.
Reply: Thank you for these kind remarks. We agree that this topic is highly interesting and will be extensively utilized in future studies. Larger-scale studies based on the proof-of-concept study results presented here are already planned and funded.
Specific comments:
- Due to young people's higher acceptance of smart phones and other reasons, the study is more inclined to target young people. The authors are also aware of this limitation in the article.
Reply: Thank you for this comment. We absolutely agree that age related availability of wearable technology is one of the main limitations of this technique. We have adapted our limitations section to better reflect this.
We still believe that this technique will see a steep increase of use as already several studies are planned and funded, but also the technology distribution in the elderly population and the generations below are bound to change over time in favor of this technique, facilitating its distribution and applicability.
- Although the operating systems of Apple and Google provided similar curved shapes to each other, it is still recommended to use the same equipment and system for testing. The authors also do not go far enough to prove whether Apple and Google are similar in terms of how they operate.
Reply: We fully understand the comment and associated concern. Certainly comparability between operating systems, or even phones from different generations by the same manufacturer is limited (i.e. due to software version, different hardware used). We have elaborated more on this in the adapted limitations section.
However, this limitation is also an advantage of the technique, as the analysis of the recovery process and activity is based on the individual data stream, thus ensuring in-person-device continuity and comparability to pre injury data. This would not be possible with a dedicated device only handed out after the injury/study start. It is both an advantage and a limitation of the technique - we have adapted the discussion and limitations section to reflect this more clearly.
Future larger scale studies will help to better the understanding of this.
Round 2
Reviewer 1 Report
Accept in present form
Reviewer 2 Report
Although there are still limitations of this study, it still has some significance for the field.